# Exact Primary Tumor Location in mCRC: Prognostic Value and Predictive Impact on Anti-EGFR mAb Efficacy

**DOI:** 10.3390/cancers14030526

**Published:** 2022-01-21

**Authors:** Annabel H. S. Alig, Volker Heinemann, Michael Geissler, Ludwig Fischer von Weikersthal, Thomas Decker, Kathrin Heinrich, Swantje Held, Lena Weiss, Laura E. Fischer, Nicolas Moosmann, Arndt Stahler, Ivan Jelas, Annika Kurreck, Jobst C. von Einem, Anke C. Reinacher-Schick, Andrea Tannapfel, Clemens Giessen-Jung, Sebastian Stintzing, Dominik P. Modest

**Affiliations:** 1Department of Hematology, Oncology, and Tumorimmunology, Charité—Universitätsmedizin Berlin, Corporate Member of Freie Universität Berlin and HumboldtUniversität zu Berlin, 10117 Berlin, Germany; annabel.alig@charite.de (A.H.S.A.); arndt.stahler@charite.de (A.S.); ivan.jelas@charite.de (I.J.); annika.kurreck@charite.de (A.K.); jobst.von-einem@charite.de (J.C.v.E.); sebastian.stintzing@charite.de (S.S.); 2Department of Medical Oncology & Comprehensive Cancer Center, University Hospital Grosshadern, Ludwig Maximilians Universität (LMU), 81377 Munich, Germany; volker.heinemann@med.uni-muenchen.de (V.H.); kathrin.heinrich@med.uni-muenchen.de (K.H.); lena.weiss@med.uni-muenchen.de (L.W.); laura.fischer@med.uni-muenchen.de (L.E.F.); clemens.giessen@med.uni-muenchen.de (C.G.-J.); 3German Cancer Consortium (DKTK), DKFZ, 69120 Heidelberg, Germany; 4Hospital Karlsruhe, 76133 Karlsruhe, Germany; michael.geissler@klinikum-karlsruhe.de; 5Klinikum St. Marien, 92224 Amberg, Germany; weikersthal.ludwig@klinikum-amberg.de; 6Onkologische Praxis, 88212 Ravensburg, Germany; decker@onkonet.eu; 7ClinAssess GmbH, 51379 Leverkusen, Germany; s.held@clinassess.de; 8Krankenhaus Barmherzige Brüder Regensburg, 93049 Regensburg, Germany; nicolas.moosmann@barmherzige-regensburg.de; 9St. Josef-Hospital, Klinikum der Ruhr-Universität Bochum, 44791 Bochum, Germany; anke.reinacher@rub.de; 10Pathologisches Institut der Ruhr Universität Bochum, 44789 Bochum, Germany; andrea.tannapfel@pathologie-bochum.de

**Keywords:** anti-EGFR antibody, metastatic colorectal cancer, primary tumor location, *RAS/BRAF* wild-type

## Abstract

**Simple Summary:**

Sidedness of primary tumor is a well-established prognostic marker and is predictive for anti-EGFR efficacy in *RAS/BRAF* wild-type metastatic colorectal cancer (mCRC) patients. As molecular markers change rather continuously throughout the colon, we ask whether the exact primary tumor location (PTL) is a better prognostic marker than sidedness and predictive for anti-EGFR efficacy in *RAS/BRAF* wild-type mCRC. We retrospectively analyzed five studies containing various therapy protocols concerning primary tumor location, dividing the colorectal frame into six segments. In our cohort, PTL has a prognostic impact on disease spread and overall survival. Only distal segments benefitted when receiving anti-EGFR containing therapy regarding overall survival. Intermediate segments were indifferent and caecal primaries had a detrimental effect receiving anti-EGFR based therapy. Being a retrospective analysis and challenging the standard of basing anti-EGFR treatment on sidedness in *RAS/BRAF* wild-type mCRC, future studies are necessary to confirm and further investigate our hypothesis-generating results.

**Abstract:**

Primary tumor sidedness (left vs. right) has prognostic and predictive impact on anti-EGFR agent efficacy and thus management of metastatic colorectal cancer (mCRC). This analysis evaluates the relevance of primary tumor location (PTL) in *RAS/BRAF* wild-type mCRC patients, when dividing the colorectal frame into six segments. This pooled analysis, performed on a single-patient basis of five randomized first-line therapy trials, evaluates the impact of exact PTL classification on baseline characteristics, prognosis and prediction of anti-EGFR antibody efficacy by chi-square and log-rank tests, the Kaplan–Meier method, Cox and logistic regressions. The PTL was significantly associated with metastatic spread: liver (*p* = 0.001), lung (*p* = 0.047), peritoneal (*p* < 0.001) and lymph nodes (*p* = 0.048). A multivariate analysis indicated an impact on anti-EGFR agent efficacy in terms of overall survival depending on the exact primary tumor location: from detrimental in caecal (HR 2.63), rather neutral effects in the ascending colon (HR 1.24), right flexure/transverse colon (HR 0.99) and left flexure/descending colon (HR 0.91) to clear benefit in sigmoid (HR 0.71) and rectal (HR 0.58) primaries. Exact primary tumor location affects anti-EGFR antibody efficacy in a rather continuous than a dichotomous fashion in *RAS/BRAF* wild-type mCRC patients. This perspective might help to support clinical decisions when anti-EGFR antibodies are considered.

## 1. Introduction

Primary tumor location (PTL), usually defined as left vs. right sidedness with a cut-off at the splenic flexure, is a prognostic and predictive biomarker in the treatment of metastatic colorectal cancer (mCRC) [1,2,3,4]. Patients presenting with right-sided mCRC have a dismal prognosis compared to patients with left-sided mCRC. Additionally, the PTL also predicts the efficacy of antibodies targeting the epidermal growth factor receptor (EGFR) in *RAS* wild-type tumors: patients presenting with left-sided mCRC (unlike right-sided mCRC) derive a substantial benefit in overall survival (OS) with anti-EGFR antibody therapy [1,2]. The role of anti-EGFR antibodies in right-sided *RAS* wild-type mCRC appears conflicting as objective responses could still be improved [1,2]. However, given the lack of survival benefit, the use of anti-EGFR antibodies is reserved to left-sided mCRC with *RAS* and potentially *BRAF* wild-type tumors [1,2,5].

Numerous efforts have been made to find a clear molecular correlate of clinical observations that establish PTL as a tool for clinical decision making. Although various molecular factors—occurring with differing frequencies throughout the colon and rectum—can be described [6,7,8,9,10], a usable classifier still has to be established. A key problem in the development of a molecular-based “right vs. left mCRC” classifier might be that clear cut-offs cannot be found for most molecular markers and that the biology of molecular differences throughout the colorectum underlies rather continuous changes than dichotomous distributions [10,11].

Therefore, this pooled analysis of five randomized trials evaluates the impact of the location of the primary tumor in patients with mCRC, breaking up the dichotomy of left- vs. right-sided colorectal cancer into six subgroups (i.e., caecum, ascending colon, right flexure plus transverse colon, left flexure plus descending colon, sigmoid and rectum) better acknowledging the continuum hypothesis [11].

We ask the question to which extent the PTL influences disease characteristics (in terms of metastatic spread and patient characteristics), the prognosis in patients with *RAS/BRAF* wild-type tumors and predicts EGFR-antibody efficacy.

As this investigation is a retrospective and unplanned analysis of five studies containing and comparing different treatment strategies, our findings should be interpreted as hypothesis generating.

## 2. Materials and Methods

### 2.1. Trials

We performed a retrospective analysis of five trials addressing patients with previously untreated mCRC (FIRE-1, CIOX, FIRE-3, XELAVIRI, VOLFI). Reports of the trials have been published previously [12,13,14,15,16,17,18,19,20,21,22]. All trials were conducted according to the Declaration of Helsinki and were approved by ethics committees. Table 1 provides an overview of the studies.

### 2.2. Patients

An anonymized clinical database of *RAS/BRAF* wild-type mCRC patients of the trials was established including the following information for each patient: trial, treatment arm, use of EGFR-antibody, age, sex, ECOG performance status, tumor characteristics (primary tumor site, metastatic sites) and prior adjuvant treatment. Tumor samples assigned to each patient had been tested for *RAS*- and *BRAF V600E*-mutations as described previously [14,15,16,17,18,19,20,21,22]: all tumor samples were assessed for mutations in *BRAF* exon 15 and *KRAS* exon 2 (codon 12 and 13). Except for those of the CIOX study, each tumor sample was tested for *KRAS* and *NRAS* exon 3 (codon 61) and exon 4 (codon 146) mutations. The tumor samples of FIRE-3, VOLFI and XELAVIRI were further assessed for mutations of *NRAS* and *KRAS* exon 3 codon 59 and exon 4 codon 117. Microsatellite status was obtained of all samples coming from FIRE-3.

### 2.3. Primary Tumor Location

Information on exact primary tumor location was extracted from the respective study report forms and differentiated into caecum, ascending colon, right flexure, transverse colon, left flexure, descending colon, sigmoid and rectum. Due to small numbers, tumors of the right flexure and transverse colon were analyzed as one group as well as tumors originating from the left flexure and the descending colon. Patients with more than one primary in more than one of these segments were excluded from the analysis of PTL. Patients with tumors of the recto-sigmoid region were analyzed as sigmoid carcinoma.

### 2.4. Treatment

Treatment procedures were described in detail in the previous publications and are summarized in Table 1. Details concerning dosages are displayed in Table 2. For further analysis, treatments were categorized as anti-EGFR based or not.

### 2.5. Definition of Overall Survival and Objective Response Rate

Progression-free survival (PFS) was defined as time from randomization to first progression of disease or death from any cause (whatever occurred first). In addition to this classic definition, a PFS event in the XELAVIRI trial was also defined as use of a new anticancer drug. OS was defined as time from randomization to death from any cause. Objective response rate (ORR) was evaluated according to classifications of the WHO (FIRE-1), RECIST 1.0 (CIOX, FIRE-3) or RECIST 1.1 (XELAVIRI, VOLFI).

### 2.6. Statistical Analysis

All statistical analyses were performed using SAS software (version 9.4; SAS Institute, Cary, NC) and SPSS version 25.0 software (IBM Corporation, Amonk, NY, USA). Baseline characteristics were compared using chi-square tests. Survival was expressed as medians including 95% confidence intervals (95% CI) by Kaplan–Meier method and compared using log-rank tests. Univariate and multivariate Cox regression analyses were used to conduct subgroup analyses. The two-sided significance level for uni- and multivariate analysis was set to 0.05 and estimates are reported with 95% CI. Patients with missing data were excluded of the respective analysis. Multivariate analyses of progression-free survival and overall survival included characteristics that were reported in all studies: study, sex, age, ECOG performance status, liver-limited disease, presence of peritoneal metastasis and prior adjuvant chemotherapy. Multivariate analyses of objective response rate included: sex, age, ECOG performance status, liver-limited disease, presence of peritoneal metastasis and prior adjuvant chemotherapy.

## 3. Results

### 3.1. Patient and Tumor Characteristics

The study population of all five studies consisted of a total of 1908 patients. Of 1809 patients, exact primary tumor location (PTL) was known. Molecular characteristics were known for 1333 patients: 717 patients with *RAS/BRAF* wild-type tumors, 514 patients with *RAS* mutant tumors, 102 patients with *BRAF V600E* mutant tumors. The population analyzed in this article comprises 717 patients with known exact primary tumor location and with *RAS/BRAF* wild-type tumors. Please also refer to Appendix A.

An overview of contributing studies and frequency of each location is shown in Appendix A. Differences in terms of patients’ and tumor characteristics according to exact primary tumor location were evident mostly in terms of disease spread and affected presence of liver metastases (*p* = 0.001), liver-limited disease (*p* = 0.029), presence of lung metastases (*p* = 0.047) and peritoneal lesions (*p* < 0.001). In addition, the number of organs involved was influenced by PTL (*p* = 0.031), with rectal cancers being associated with the numerically lowest frequency of one-organ disease (38.4%) and tumors of the splenic flexure and the descending colon being associated with the highest frequency of one organ disease (57.4%). Caecal primary tumors were less frequently associated with liver-limited disease (30.0%) and more frequently associated with peritoneal metastases (20.0%) as compared with other PTL.

Detailed baseline characteristics of patients and tumors are summarized in Table 3A–C. A graphical overview of correlations of baseline and tumor characteristics of interest with significant differences according to PTL is provided as Appendix A.

### 3.2. Objective Response Rates According to Primary Tumor Location

Generally, irrespective of the exact therapy, objective response rates (ORR) were around 60% in all primary tumor locations analyzed in our cohort, ranging from 57% to 67%. No significant difference according to PTL was evident (*p* = 0.784, Χ^2^-Test), although numerically higher response rates in left-sided primary tumors were observed (60–67%). Details are summarized in Appendix A.

### 3.3. Prognostic Impact of Primary Tumor Location

In the analyzed population, stratification of outcome by exact PTL introduced significant differences in terms of overall survival (*p* = 0.003), but not in progression-free survival (*p* = 0.132). The numerically longest PFS and OS were observed in patients with primary tumors located in the left flexure/descending colon (OS 33.1 months, PFS 11.5 months) and sigmoid (OS 31.0 months, PFS 10.7 months). Interestingly, rectal cancers were associated with slightly less favorable outcomes with a median OS of 28.2 months as compared to the other left-sided primary tumor locations (L-PTL). Numerically shortest survival trended to occur in primary tumors of the caecum (OS 20.5 months) and the ascending colon (OS 18.3 months) although no significant difference was seen in between right-sided PTL (R-PTL).

Kaplan–Meier curves for all patients are indicated in Figure 1.

### 3.4. Predictive Impact on Anti-EGFR Antibody Efficacy of Primary Tumor Location

ORR appeared to be influenced by PTL with left-sided locations being associated with a higher chance to achieve objective response to treatment when receiving anti-EGFR targeted therapy. However, the only segment with a clear signal of improved objective response was sigmoid mCRC. Of note, in right-sided locations, no clear or homogeneous signal of either detriment or benefit in terms of objective response was detected. This assessment was confirmed through uni- and multivariate analysis: even though there trended to be a benefit of anti-EGFR treatment regarding ORR in sigmoid PTL in the multivariate analysis (HR 1.68), no substantial trending in favor or against anti-EGFR containing treatment was seen.

In terms of PFS, a detrimental effect of anti-EGFR antibodies in patients with caecal *RAS/BRAF* wild-type mCRC was observed (multivariate Cox regression: HR 2.50), whereas no significant differences in all other segments were evident with rather neutral effects evolving with more distal primary tumor location.

However, OS appeared to improve with anti-EGFR antibodies the further distal the primary tumor was located. Whereas in patients with caecal primaries the use of anti-EGFR antibodies resulted in a significant detrimental effect (multivariate Cox regression: HR 2.63), neutral efficacy was seen in the segments in between the caecum and the sigmoid. The subgroups demonstrating a clear and statistically significant benefit with anti-EGFR antibody treatment in multivariate analysis were patients with primaries located in the sigmoid (HR 0.71; 95% CI 0.53–0.95) and especially in the rectum (HR 0.58; 95% CI 0.43–0.77). An overview of the predictive effect of each PTL on anti-EGFR antibody efficacy is provided as Figure 2 (Kaplan–Meier estimates) and Figure 3 (Cox regression).

## 4. Discussion

The presented analysis based on five randomized trials including data of 717 patients with *RAS/BRAF* wild-type tumors represents a large and robust basis to evaluate the impact of exact primary tumor location in *RAS/BRAF* wild-type mCRC on clinical characteristics, prognosis and prediction of anti-EGFR antibody efficacy, when PTL is divided into six segments.

In our cohort of *RAS/BRAF* wild-type mCRC, PTL significantly correlated with disease spread, both in terms of involved organs as well as in terms of number of organs involved. Of interest and clinical relevance might be the high frequency of liver-limited disease in patients with mCRC arising from the splenic flexure and descending colon providing a favorable precondition for interdisciplinary management and consecutively good outcomes.

Whereas the usual differentiation between left- vs. right-sided colorectal cancer is a known prognostic factor of OS in mCRC [2,3,4], our analysis suggests that certain differences between the individual segments could exist. Of interest, the shortest OS was detected in patients with mCRC deriving from primaries of the ascending colon.

Amongst patients with L-PTL, differences in outcome were not significant. Of note, rectal primary tumors appeared to be associated with the numerically shortest OS within the left-sided primaries. This finding might compare favorably to newly published data that report rectal tumors to have both characteristics of left-sided and right-sided colon cancer [23].

Clear effects of PTL on OS were observed in patients with *RAS/BRAF* wild-type mCRC with regard to the use of anti-EGFR antibodies. Interestingly, patients with primary tumors of the caecum derived a substantial disadvantage with anti-EGFR therapy, whereas substantial benefit was seen in patients with rectal cancers and—to a lesser extent —in patients with sigmoid primaries and rather neutral effects were seen in between.

Taken together, the data suggest a rather continuous increase in benefit from anti- EGFR antibodies from proximal (right) to distal (left) segments rather than the currently established dichotomous perspective of left (benefit from anti-EGFR) vs. right (no benefit from anti-EGFR) mCRC [1,2]. This perspective is supported by the molecular analysis of colorectal tumors that rather proposes a continuum of biological changes from proximal to distal segments of colon and rectum [10,11]. Based on these findings, the treatment of rectal and sigmoid *RAS/BRAF* wild-type mCRC with anti-EGFR antibodies appears a clinical necessity. Particularly, mCRC patients with rectal primaries may achieve an enormous advantage with anti-EGFR targeted therapy, while patients with mCRC originating from sigmoid primaries also achieve a clinically relevant benefit. For patients with left-sided primary tumors in the segments between left flexure and descending colon, our results raise the question of if and to which extent patients in this subgroup benefit from anti-EGFR based therapy. Naturally, given the limited sample sizes, this particular question of anti-EGFR based treatment benefit can also be asked with regard to further “intermediate” segments (i.e., transverse colon) [24].

The effect of anti-EGFR antibodies did not impact on PFS in our cohort, illustrating—in accordance with previous publications—that PFS does not necessarily reflect the efficacy of anti-EGFR antibodies [21,25,26,27] and these drugs may rather impact response related endpoints and OS [16,21,25,28,29].

Even though larger analyses suggest a higher ORR even on right-sided tumors during anti-EGFR mAb treatment [1,2], our analysis demonstrated no improvement of objective response rate with anti-EGFR mAb.

Challenging the idea of dichotomous left vs. right classification in mCRC, our hypothesis of a rather continuous increase in anti-EGFR therapy related benefit from proximal to distal segments of the colorectum as observed in our analyses clearly supports the idea that the underlying molecular equivalent is likely a non-dichotomous biomarker (combination), potentially including EGFR-ligands, HER-3 messenger RNA or miR-31-3p [30,31,32,33,34]. Interestingly, there was a clear detrimental effect on efficacy of EGFR-targeted antibodies in caecal PTL. This observation is clearly less pronounced in more distal parts of R-PTL (i.e., ascending to transverse colon) and may suggest that right-sided mCRC is a heterogeneous group of patients and the key driver of detrimental effects could be caecal primaries.

Our pooled analysis is limited by its retrospective and exploratory design as well as by the small patient numbers in some segments of the colon with an overall low frequency of primary tumors. Furthermore, a pooled analysis of five studies (of those two directly randomizing anti-EGFR mAb use vs. no mAb or anti-VEGF mAb) using various treatment regimens and therefore, containing heterogeneous populations may have invoked potential undetected biases. Furthermore, additional information, including exact stage (size of primary tumor, nodal status), grade and morphological subtype (e.g., well or poorly differentiated tumors), is unavailable and a potential uneven distribution of these features between segments may have biased our observation. Additionally, microsatellite stability status was not characterized in all cohorts and the absence of this biomarker may have led to some small bias. As the frequency of MSI-high tumors in mCRC is less than 5% and the effect of anti-EGFR targeted therapy is unclear in these tumors, the effect of a possible bias is unknown and might be quite small [23,24,25,26]. In addition, a characterization of consensus molecular subgroups may have helped to elucidate potential subgroups within the intermediate segments of the colon (despite also reducing the sample sizes in analyzed segments substantially), but is not available for all studies [9].

## 5. Conclusions

Exact primary tumor location is associated with various aspects of disease spread; in particular, involved organs appear to depend on the primary tumor location—even in context with a molecular selection of rather favorable *RAS/BRAF* wild-type disease. Furthermore, exact primary tumor location is associated with a rather continuous increase in anti-EGFR antibody efficacy from proximal to distal segments of the colorectum in patients with *RAS/BRAF* wild-type tumors. This finding challenges the current use of the dichotomous left vs. right classifier to choose a first-line treatment and may provide a basis for differentiated clinical decision making.

Future studies are necessary to confirm our hypothesis-generating results and to further analyze the rather neutral effects of anti-EGFR treatment in the segments between the right flexure and the descending colon. Within this population, subgroups with benefit from anti-EGFR treatment might be found if larger cohorts were available. Additional characterization of tumors might help to improve the selection of patients likely to achieve clinical benefit when treated with an anti-EGFR antibody.

## Figures and Tables

**Figure 1 cancers-14-00526-f001:**
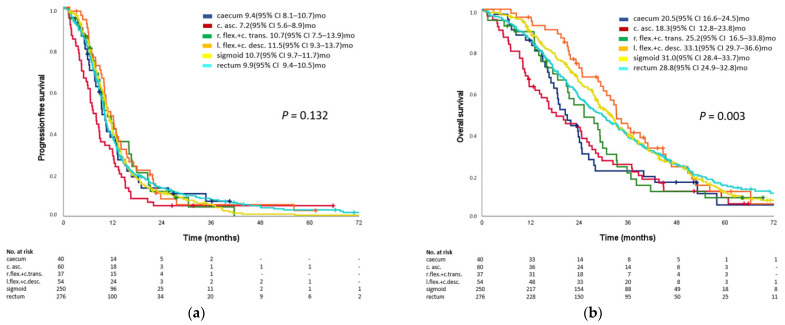
**Kaplan-Meier estimates of efficacy endpoints**. (**a**) Kaplan–Meier estimates of progression -free survival and overall survival according to exact PTL in RAS/BRAF wild-type tumors. (**b**) Kaplan–Meier estimates of overall survival according to exact PTL in RAS/BRAF wild-type tumors. Legend: 95% CI = confidence interval; c. asc. = ascending colon; c. desc. = descending colon; c. trans = transverse colon; l. flex/l. fl. = left flexure; r. flex./r. fl. = right flexure; no. = number; OS = overall survival; PFS = progression-free survival. *p*-values: log-rank test.

**Figure 2 cancers-14-00526-f002:**
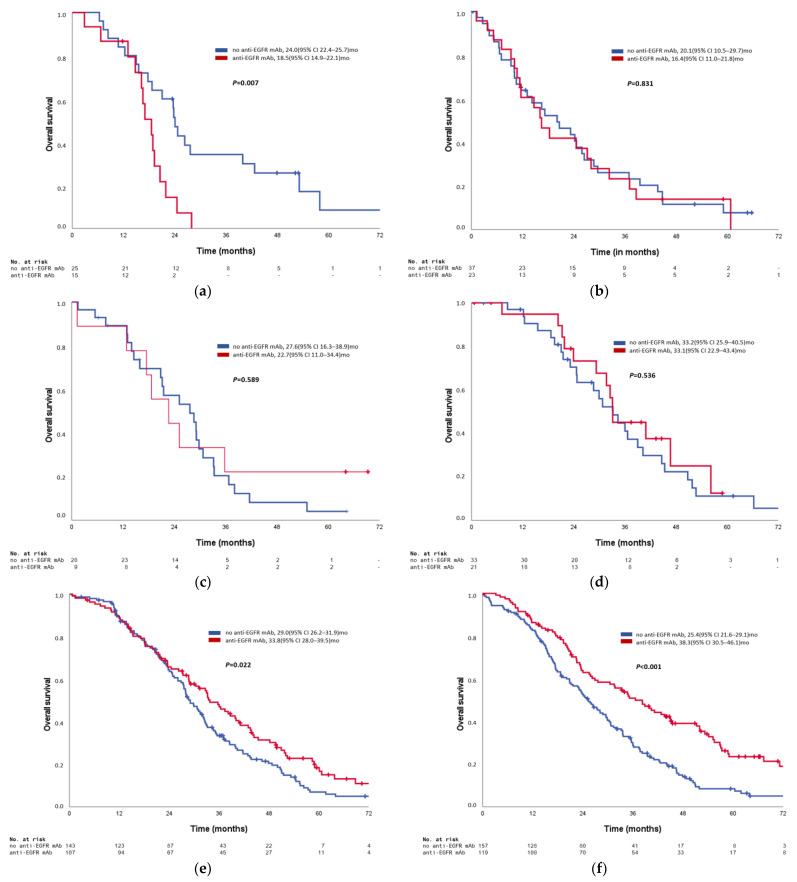
**Kaplan**–**Meier estimates of overall survival according to exact primary tumor location**. (**a**) Kaplan–Meier estimates of overall survival in RAS/BRAF wild-type mCRC with primary tumor location in the caecum according to anti-EGFR use. (**b**) Kaplan–Meier estimates of overall survival in RAS/BRAF wild-type mCRC with primary tumor location in the ascending colon according to anti-EGFR use. (**c**) Kaplan–Meier estimates of overall survival in RAS/BRAF wild-type mCRC with primary tumor location in the right flexure/transverse colon according to anti-EGFR use. (**d**) Kaplan–Meier estimates of overall survival in RAS/BRAF wild-type mCRC with primary tumor location in the left flexure/descending colon according to anti-EGFR use. (**e**) Kaplan–Meier estimates of overall survival in RAS/BRAF wild-type mCRC with primary tumor location in the sigmoid according to anti-EGFR use. (**f**) Kaplan–Meier estimates of overall survival in RAS/BRAF wild-type mCRC with primary tumor location in the rectum according to anti-EGFR use. Legend: 95% CI = 95% confidence interval; c. asc. = ascending colon c. desc.=descending colon;; c. trans = transverse colon; l. flex/l. fl. = left flexure; mCRC = metastatic colorectal cancer; no. = number; r. flex./r. fl. =right flexure; OS = overall survival. *p*-values: log rank test.

**Figure 3 cancers-14-00526-f003:**
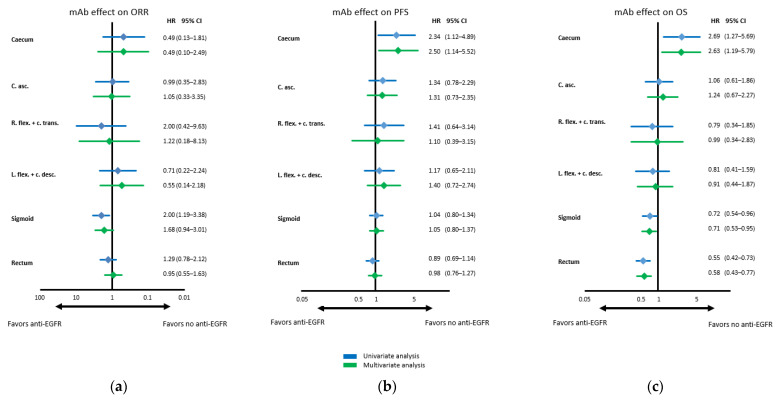
**Forest plots of estimated anti-EGFR effect**. (**a**) Forest plots of estimated anti-EGFR effect on objective response rate according to exact primary tumor location in RAS/BRAF wild-type tumors. (**b**) Forest plots of estimated anti-EGFR effect on progression-free survival according to exact primary tumor location in RAS/BRAF wild-type tumors. (**c**) Forest plots of estimated anti-EGFR effect on overall survival according to exact primary tumor location in RAS/BRAF wild-type tumors. Legend: Uni- and multivariate analyses of anti-EGFR effect. Results of univariate analysis are depicted as blue bar/diamonds and results of multivariate analysis as green bar/diamonds. Multivariate analyses of progression-free survival and overall survival included study, sex, age, ECOG performance status, liver-limited disease, peritoneal metastasis and prior adjuvant chemotherapy as factors. Multivariate analyses of objective response rate included sex, age, ECOG performance status, liver-limited disease, peritoneal metastasis and prior adjuvant chemotherapy as factors. 95% CI = 95% confidence interval; c. asc. = ascending colon; c. desc. = descending colon; c. trans.= transverse colon; HR = hazard ratio; l. flex. = left flexure; mAb = monoclonal antibody; ORR = objective response rate; OS = overall survival; PFS = progression-free survival; r. flex. = right flexure.

**Table 1 cancers-14-00526-t001:** Characteristics of the five studies included in the pooled analysis.

	FIRE-1	CIOX	FIRE-3	XELAVIRI	VOLFI
**Phase of study**	III	II	III	III	II
**Country**	Germany	Germany	Germany, Austria	Germany	Germany
**No. of centers**	48	35	110, 6	82	21
**Recruiting period**	07/2000–10/2004	09/2004–12/2006	01/2007–09/2012	12/2010–04/2016	06/2011–01/2016
**Primary endpoint**	PFS	ORR	ORR	TFS	ORR
**OS censored in database**	12.9%	19.8%	12.7%	15.2%	34.4% *
**Treatment arms**					
Arm A	FUFIRI	CAPIRI + Cet	FOLFIRI + Cet	FP + Bev –> PD –> FP + Iri + Bev	mFOLFOXIRI + Pani
Arm B	mIROX	CAPOX + Cet	FOLFIRI + Bev	FP + Iri + Bev	FOLFOXIRI
**Previous adj.** **chemotherapy allowed**	yes (no TOP1 inh., no platinum)	yes (no TOP1 inh.)	yes	yes	yes
**Required time between adj. chemotherapy and relapse**	6 months	6 months	6 months	6 months	6 months
**RECIST version**	- (WHO)	1.0	1.0	1.1	1.1
**Trial finder registration**	-	NCT00254137	NCT004 33927	NCT012 49638	NCT013 28171
**Eligibility criteria**					
Age (in years)	18–75	18–75	18–75	≥18	≥18
ECOG	-	-	≤2	≤1	≤1
Karnowsky	≥70%	≥70%	-	-	-

**Legend:** adj. = adjuvant; TOP1 inh. = topoisomerase 1 inhibitors; FUFIRI = irinotecan, leucovorin and infusional fluorouracil; Iri = irinotecan; mIROX = irinotecan and oxaliplatin; Cet = cetuximab; CAPIRI = oral fluorouracil and irinotecan; CAPOX = oral fluorouracil and oxaliplatin; FOLFIRI = infusional fluorouracil, leucovorin, and irinotecan; FOLFOX = infusional fluorouracil, leucovorin, and oxaliplatin; Bev = bevacizumab; FP = fluorouracil (oral or infusional, infusional with leucovorin); mFOLFOXIRI = modified FOLFOXIRI; FOLFOXIRI = infusional fluorouracil, leucovorin, oxaliplatin and irinotecan; No.=number; Pani=Panitumumab;ORR = objective response rate; TFS = time to failure of strategy; OS = overall survival; PFS = progression-free survival; PD = progressive disease;WT = wild-type. * Database not final for OS.

**Table 2 cancers-14-00526-t002:** Treatments protocols according to each study.

		Induction	Escalation
**XELAVIRI**	**Sequential therapy arm**	**CAP + Bev (q3w)** CAP 1250 mg/m^2^ 1-0-1 p.o.; days 1–14 + Bev 7.5 mg/kg/bw i.v.; day 1	**CAPIRI + Bev (q3w)** CAP 800 mg/m^2^ 1-0-1 p.o.; days 1–14 + Iri 200 mg/m^2^ i.v.; day 1 + Bev 7.5 mg/kg/bw i.v.; day 1
		**FUFA + Bev (q2w)** FA 400 mg/m^2^ i.v.; day 1 + 5-FU bolus 400 mg/m^2^ i.v.; day 1 + 5-FU 2400 mg/m^2^ over 46 h i.v.; starting day 1 + Bev 5 mg/kg/bw i.v.; day 1	**FOLFIRI + Bev (q2w)** Iri 180 mg/m^2^ i.v.; day 1 + FA 400 mg/m^2^ i.v.; day 1 + 5-FU bolus 400 mg/m^2^ i.v.; day 1 + 5-FU 2400 mg/m^2^ over 46h i.v.; starting day 1 + Bev 5 mg/kg/bw i.v.; day 1
	**Combination therapy arm**	**Induction**	**Intermittent de-escalation** (in case of at least stable disease for more than six months)
		**CAPIRI + Bev (q3w)** or **FOLFIRI + Bev (q2w)**	**CAP + Bev (q3w)** or **FUFA + Bev (q2w)**
**FIRE-1**	**Arm A**	**FUFIRI (q7w)** Iri 80 mg/m^2^ i.v.; day 1, 8, 15, 22, 29, 36, 43 + FA 500 mg/m^2^ i.v.; day 1, 8, 15, 22, 29, 36, 43 + 5-FU 2000 mg/m^2^ over 24 h i.v.; starting day 1, 8, 15, 22, 29, 36, 43	
	**Arm B**	**mIROX (q7w)** Iri 80 mg/m^2^ i.v.; day 1, 8, 15, 22, 29, 36, 43 + Ox 85 mg/m^2^ i.v.; days 1, 15, 29	
**CIOX**	**Arm A**	**CAPIRI + Cet (q3w)** CAPIRI + Cet 400 mg/m^2^ i.v. on day 1 of c1 or Cet 250 mg/m^2^ i.v.; day 8, 15; starting c2 day 1	
	**Arm B**	**CAPOX + Cet (q3w)** CAP 1000 mg/m^2^ 1-0-1 p.o.; days 1–14 + Ox 130 mg/m^2^ i.v.; day 1 + Cet 400 mg/m^2^ i.v. on day 1 of c1 or Cet 250 mg/m^2^ i.v.; day 8, 15; starting c2 day 1	
**FIRE-3**	**Arm A**	**FOLFIRI + Cet (q2w)** FOLFIRI + Cet 400 mg/m^2^ i.v. on day 1 of c1 or Cet 250 mg/m^2^ i.v.; day 8; starting c2 additionally on day 1	
	**Arm B**	**FOLFIRI + Bev (q2w)**	
**VOLFI**	**Arm A**	**mFOLFOXIRI + Pani (q2w)** Pani 6 mg/kg/bw i.v.; day 1 + Iri 150 mg/m^2^ i.v.; day 1 + Ox 85 mg/m^2^ i.v.; day 1 + FA 200 mg/m^2^ i.v.; day 1 + 5-FU 3000 mg/m^2^ over 48h i.v.; starting day 1	
	**Arm B**	**FOLFOXIRI (q2w)** Iri 165 mg/m^2^ i.v.; day 1 + Ox 85 mg/m^2^ i.v.; day 1 + FA 200 mg/m^2^ i.v.; day 1 + 5-FU 3200 mg/m^2^ over 48 h i.v.; starting day 1	

**Legend:** Bev = Bevacizumab; bw = body weight; c = cycle; CAP = capecitabine; CAPIRI = capecitabine and irinotecan; CAPOX = capecitabine and oxaliplatin; Cet = Cetuximab; FA = racemic folinic acid/leucovorin; 5-FU = 5-fluorouracil; FUFA = 5-FU and leucovorin; FOLFIRI = infusional fluorouracil, leucovorin, and irinotecan; FOLFOX = infusional fluorouracil, leucovorin and oxaliplatin; FOLFOXIRI = infusional fluorouracil, leucovorin, oxaliplatin and irinotecan; mFOLFOXIRI = modified FOLFOXIRI; FUFIRI = irinotecan, leucovorin and infusional fluorouracil; Iri = irinotecan; mIROX = irinotecan and oxaliplatin; i.v. = intravenous; OX = oxaliplatin; Pani = Panitumumab; p.o. = per os.

**Table 3 cancers-14-00526-t003:** Tumor characteristics.

A. Tumor characteristics: *RAS/BRAF* wild-type tumors, n = 717.						
**Characteristics**	**Caecum** **(n = 40)**	**C. asc.** **(n = 60)**	**R. flex. +** **c. trans.** **(n = 37)**	**L. flex. +** **c. desc.** **(n = 54)**	**Sigmoid** **(n = 250)**	**Rectum** **(n = 276)**
**Study**						
FIRE-1	6 (15.0%)	7 (11.7%)	2 (5.4%)	11 (20.4%)	31 (12.4%)	30 (10.9%)
CIOX	6 (15.0%)	6 (10.0%)	4 (10.8%)	3 (5.6%)	29 (11.6%)	31 (11.2%)
FIRE-3	16 (40.0%)	32 (53.3%)	17 (50.0%)	30 (55.6%)	125 (50.0%)	130 (47.1%)
XELAVIRI	7 (17.5%)	15 (25.0%)	13 (35.1%)	8 (14.8%)	39 (12.4%)	60 (21.7%)
VOLFI	5 (12.5%)	0 (0.0%)	1 (2.7%)	2 (3.7%)	26 (10.4%)	25 (9.1%)
**Antibody**						
none	8 (20.0%)	7 (11.7%)	3 (8.1%)	11 (20.4%)	41 (16.4%)	35 (12.7%)
Anti-EGFR	15 (37.5%)	23 (38.3%)	9 (24.3%)	21 (38.9%)	107 (42.8%)	119 (43.1%)
Anti-VEGF	17 (42.5%)	30 (50.0%)	25 (67.6%)	22 (40.7%)	102 (40.8%)	122 (44.2%)
**Sex**						
Male	25 (62.5%)	42 (70.0%)	23 (62.2%)	35 (64.8%)	182 (72.8%)	210 (76.1%)
Female	15 (37.5%)	18 (30.0%)	14 (37.8%)	19 (35.2%)	68 (27.2%)	66 (23.9%)
**Age** (in years)						
≤60	11 (31.4%)	20 (33.3%)	5 (13.9%)	15 (28.8%)	76 (33.9%)	89 (35.5%)
>60–≤70	10 (28.6%)	20 (33.3%)	18 (50.0%)	22 (42.3%)	91 (40.6%)	101 (40.2%)
>70	14 (40.0%)	20 (33.3%)	13 (36.1%)	15 (28.8%)	57 (25.4%)	61 (24.3%)
**ECOG**						
0	23 (57.5%)	28 (46.7%)	20 (54.1%)	35 (64.8%)	156(62.9%)	180 (65.2%)
≥1	17 (42.5%)	32 (53.3%)	17 (45.9%)	19 (35.2%)	92 (36.8%)	95 (34.4%)
Unknown	0 (0.0%)	0 (0.0%)	0 (0.0%)	0 (0.0%)	0 (0.0%)	1 (0.4%)
**Metastatic spread**						
Liver	33 (82.5%)	50 (83.3%)	36 (97.3%)	50 (92.6%)	225 (90.0%)	216 (78.3%)
Liver-limited	12 (30.0%)	20 (33.3%)	15 (40.5%)	29 (53.7%)	99 (39.6%)	86 (31.2%)
Lung	12 (30.0%)	23 (38.3%)	11 (29.7%)	13 (24.1%)	75 (30.0%)	113 (40.9%)
Peritoneum	8 (20.0%)	4 (6.7%)	4 (10.8%)	2 (3.7%)	17 (6.8%)	6 (2.2%)
**No. of metastatic sites**						
1	16 (40.0%)	26 (43.3%)	15 (40.5%)	31 (57.4%)	106(42.4%)	106 (38.4%)
≥2	19 (47.5%)	33 (55.0%)	21 (56.8%)	20 (37.0%)	117 (46.8%)	143 (51.8%)
Unknown	5 (12.5%)	1 (1.7%)	1 (2.7%)	3 (5.6%)	27 (10.8%)	27 (9.8%)
**Onset of metastases**						
Synchronous	20 (50.0%)	48 (80.0%)	19 (51.4%)	33 (61.1%)	147 (58.8%)	149 (54.0%)
Metachronous	8 (20.0%)	6 (10.0%)	13 (35.1%)	16 (29.6%)	48 (19.2%)	69 (25.0%)
Unknown	12 (30.0%)	6 (10.0%)	5 (13.5%)	5 (9.3%)	55 (22.0%)	58 (21.0%)
**Previous chemotherapy**						
No	32 (80.0%)	57 (95.0%)	31 (83.8%)	47 (87.0%)	214 (85.6%)	188 (68.1%)
Yes	8 (20.0%)	3 (5.0%)	6 (16.2%)	7 (13.0%)	36 (14.4%)	87 (31.5%)
Unknown	0 (0.0%)	0 (0.0%)	0 (0.0%)	0 (0.0%)	0 (0.0%)	1(0.4%)
B. Tumor characteristics of patients with *RAS/BRAF* wild-type tumors treated with anti-EGFR targeted therapy, n = 294.						
**Characteristics**	**Caecum** **(n = 15)**	**C. asc.** **(n = 23)**	**R. flex. +** **c. trans.** **(n = 9)**	**L. flex. +** **c. desc.** **(n = 21)**	**Sigmoid** **(n = 107)**	**Rectum** **(n = 119)**
**Study**						
CIOX	6 (40.0%)	6 (26.1%)	4 (44.4%)	3 (14.3%)	29 (27.1%)	31 (26.1%)
FIRE-3	6 (40.0%)	17 (73.9%)	5 (55.6%)	16 (76.2%)	62 (57.9%)	68 (57.1%)
VOLFI	3 (20.0%)	0 (0%)	0 (0%)	2 (9.5%)	16 (15.0%)	20 (16.8%)
**Sex**						
Male	6 (40.0%)	16 (69.6%)	6 (66.7%)	12 (57.1%)	81 (75.7%)	92 (77.3%)
Female	9 (60.0%)	7 (30.4%)	3 (33.3%)	9 (42.9%)	26 (24.3%)	27 (22.7%)
**Age** (in years)						
≤60	4 (26.7%)	6 (26.1%)	4 (44.4%)	6 (28.6%)	40 (37.4%)	43 (36.1%)
>60-≤70	6 (40.0%)	10 (43.5%)	3 (33.3%)	9 (42.9%)	31 (29.0%)	38 (31.9%)
>70	2 (13.3%)	7 (30.4%)	2 (22.2%)	4 (19.0%)	20 (18.7%)	18 (15.1%)
unknown	3 (20.0%)	0 (0.0%)	0 (0.0%)	2 (9.5%)	16 (15.0%)	20 (16.8%)
**ECOG**						
0	8 (53.3%)	9 (%)	5 (55.6%)	15 (71.4%)	69 (64.5%)	83 (69.7%)
≥1	7 (46.7%)	14 (%)	4 (44.4%)	6 (28.6%)	38 (35.5%)	36 (30.3%)
Unknown	0 (0.0%)	0 (0.0%)	0 (0.0%)	0 (0.0%)	0 (0.0%)	(0.0%)
**Metastatic spread**						
Liver-limited *	6 (40.0%)	8 (34.8%)	4 (44.4%)	14 (66.7%)	40 (37.4%)	40 (33.6%)
Liver *	14 (93.3%)	19 (82.6%)	9 (100%)	20 (95.2%)	96 (89.7%)	101(84.9%)
Lung *	3 (20.0%)	10 (43.5%)	2 (22.2%)	3 (14.3%)	33 (30.8%)	45 (37.8%)
Peritoneum *	2 (13.3%)	2 (8.7%)	1 (11.1%)	1 (4.8%)	1 (1.0%)	3 (2.5%)
**No. of metastatic sites**						
1	6 (40.0%)	11 (47.8%)	4(44.4%)	14 (66.7%)	42 (39.3%)	43 (36.1%)
≥2	6 (40.0%)	12 (52.2%)	5 (55.6%)	4 (19.0%)	48 (44.9%)	56 (47.1%)
Unknown	3 (20.0%)	0 (0.0%)	0 (0.0%)	1 (4.8%)	17 (15.9%)	20 (16.8%)
**Onset of metastases**						
Synchronous	4 (26.7%)	15 (65.2%)	4 (44.4%)	11 (52.4%)	51 (47.7%)	47 (39.5%)
Metachronous	2 (13.3%)	2 (8.7%)	1 (11.1%)	5 (23.8%)	11 (10.3%)	21 (17.6%)
Unknown	9 (40.0%)	6 (26.1%)	4 (44.4%)	5 (23.8%)	45 (42.1%)	49 (41.2%)
**Previous chemotherapy**						
No	13 (86.7%)	21 (91.3%)	8 (88.9%)	18 (85.7%)	93 (86.9%)	89 (74.8%)
Yes	2 (13.3%)	2 (8.7%)	1 (11.1%)	3 (14.3%)	14 (13.1%)	30 (25.2%)
Unknown	0 (0.0%)	0 (0.0%)	0 (0.0%)	0 (0.0%)	0 (0.0%)	0 (0.0%)
C. Tumor characteristics of patients with *RAS/BRAF* wild-type tumors not treated with anti-EGFR targeted therapy, n = 423.						
**Characteristics**	**Caecum** **(n = 25)**	**C. asc.** **(n = 37)**	**R. flex. +** **c. trans.** **(n = 28)**	**L. flex. +** **c. desc.** **(n = 28)**	**Sigmoid** **(n = 143)**	**Rectum** **(n = 157)**
**Study**						
FIRE-1	6 (24.0%)	7 (18.9%)	2 (7.1%)	11 (33.3%)	31 (21.7%)	30 (19.1%)
FIRE-3	10 (40.0%)	15 (40.5%)	12 (42.9%)	14 (42.4%)	63 (44.1%)	62 (39.5%)
XELAVIRI	7 (28.0%)	15 (40.5%)	13 (46.4%)	8 (24.2%)	39 (27.3%)	60 (38.2%)
VOLFI	2 (8.0%)	0 (0.0%)	1 (3.6%)	0 (0.0%)	10 (7.0%)	5 (3.2%)
**Sex**						
Male	19 (76.0%)	26 (70.3%)	17 (60.7%)	23 (69.7%)	101 (70.6%)	118 (75.2%)
Female	6 (24.0%)	11 (29.7%)	11 (39.3%)	10 (30.3%)	42 (29.4%)	39 (24.8%)
**Age** (in years)						
≤60	7 (28.0%)	14 (37.8%)	1 (3.6%)	9 (27.3%)	36 (25.2%)	46 (29.3%)
>60–≤70	4 (16.0%)	10 (27.0%)	15 (53.6%)	13 (39.4%)	60 (42.0%)	63 (40.1%)
>70	12 (48.0%)	13 (35.1%)	11 (39.3%)	11 (33.3%)	37 (25.9%)	43 (27.4%)
unknown	2 (8.0%)	0 (0.0%)	1 (36%)	0 (0.0%)	10 (7.0%)	5 (3.2%)
**ECOG**						
0	15 (60.0%)	19 (51.4%)	15 (53.6%)	20 (60.6%)	87 (60.8%)	97 (61.8%)
≥1	10 (40.0%)	18 (48.6%)	13 (46.4%)	13 (39.4%)	54 (37.8%)	59 (37.6%)
Unknown	0 (0.0%)	0 (0.0%)	0 (0.0%)	0 (0.0%)	2 (1.4%)	1 (0.6%)
**Metastatic spread**						
Liver-limited *	6 (24.0%)	12 (32.4%)	11 (39.3%)	15 (45.5%)	59 (41.3%)	46 (29.3%)
Liver *	19 (76.0%)	31 (83.8%)	27 (96.4%)	30 (90.9%)	129 (90.2%)	115 (73.2%)
Lung *	9 (36.0%)	13 (35.1%)	9 (32.1%)	10 (30.3%)	42 (29.4%)	68 (43.3%)
Peritoneum *	6 (24.0%)	2 (5.4%)	3 (10.7%)	1 (3.0%)	6 (4.2%)	3 (1.9%)
**No. of metastatic sites**						
1	10 (40.0%)	15 (40.5%)	11 (39.3%)	17 (51.5%)	64 (44.8%)	63 (40.1%)
≥2	13 (52.0%)	21 (56.8%)	16 (57.1%)	16 (48.5%)	69 (48.3%)	87 (55.4%)
Unknown	2 (8.0%)	1 (2.7%)	1 (3.6%)	0 (0.0%)	10 (7.0%)	7 (4.5%)
**Onset of metastases**						
Synchronous	16 (64.0%)	33 (89.2%)	23 (82.1%)	29 (87.9%)	121 (84.6%)	99 (63.1%)
Metachronous	6 (24.0%)	4 (10.8%)	5 (17.9%)	4 (12.1%)	22 (15.4%)	57 (36.3%)
Unknown	3 (12.0%)	0 (0.0%)	0 (0.0%)	0 (0.0%)	0 (0.0%)	1 (0.6%)
**Previous chemotherapy**						
No	19 (76.0%)	36 (97.3%)	23 (82.1%)	29 (87.9%)	121 (84.6%)	102 (65.0%)
Yes	6 (24.0%)	1 (2.7%)	5 (17.9%)	4 (12.1%)	22 (15.4%)	48 (30.6%)
Unknown	0 (0.0%)	0 (0.0%)	0 (0.0%)	0 (0.0%)	0 (0.0%)	7 (4.5%)

**Legend:** Underlined studies represent studies with at least one arm containing anti-EGFR targeted therapy. C. asc. = ascending colon; L. flex. + c. desc. = left flexure and descending colon; No. = number; R. flex. + c. trans = right flexure and transverse colon; Sec. = secondary. * 1 patient missing from analysis due to missing data.

## Data Availability

The data that support the findings of this study are available from the corresponding author upon reasonable request.

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
