# Peer review of "Exact Primary Tumor Location in mCRC: Prognostic Value and Predictive Impact on Anti-EGFR mAb Efficacy"

_cancers, 2022, doi:10.3390/cancers14030526_

Round 1
Reviewer 1 Report
The authors present a meta-analysis of effects of primary tumor location (PTL) on overall survival of patients with metastatic colorectal adenocarcinoma with wildtype BRAF/RAS receiving anti-EGFR therapy. They found that PTL does have a significant impact on disease spread and overall survival.
The authors find significant differences particularly in overall survival with exact location specified and a paradoxically negative response to EGFR therapy in cecal tumors. These results challenge conventional wisdom and suggests that the right-left therapy paradigm in CRC should be revisited / updated.
However, the above conclusions rely on an even distribution of grade, stage and morphological subtype amongst the various tumor locations. This data is lacking from the results and so it is difficult to say for sure that the effects are real and not a result of imbalance distribution of the above tumor characteristics. See Major Comments.
Minor comments:
37: Is this truly continuous as opposed to just a finer subdivision of an ordinal? I realize that the changes in molecular and cellular organization of the colon has been shown to be a continuum, but the subdivision of the colon into broad categories (cecum, ascending, transverse, etc) in this study is an ordinal treatment. If one could define a location in the colon reproducibly across persons using molecular markers, I think this argument could be made.
Major comments:
The authors mention that MMR status was not controlled for in this study and may have an impact, and I wholeheartedly agree with this statement.
I think, however, the bigger problem by far is the lack of controlling for morphological subtype, grade and stage of tumor. Presumably, there is an even distribution amongst the various sites, but we have no way of assessing this from the provided data. All we know is that the included patients are presumably pTx Nx M1 (e.g they have distant metastases). What if, for instance, the cecal cohort cases are enriched for patients with Grade 3, Undifferentiated, pT4 N2b M1? This is probably not the case, but it cannot be assessed based on the data provided.
Without this data, all I can say is that the findings are interesting but need further investigation.
Author Response
The authors present a meta-analysis of effects of primary tumor location (PTL) on overall survival of patients with metastatic colorectal adenocarcinoma with wildtype BRAF/RAS receiving anti-EGFR therapy. They found that PTL does have a significant impact on disease spread and overall survival.
The authors find significant differences particularly in overall survival with exact location specified and a paradoxically negative response to EGFR therapy in cecal tumors. These results challenge conventional wisdom and suggests that the right-left therapy paradigm in CRC should be revisited / updated.
However, the above conclusions rely on an even distribution of grade, stage and morphological subtype amongst the various tumor locations. This data is lacking from the results and so it is difficult to say for sure that the effects are real and not a result of imbalance distribution of the above tumor characteristics. See Major Comments.
Dear reviewer,
First of all, thank you for your kind words. We are very grateful for your time and your efforts.
Minor comments:
37: Is this truly continuous as opposed to just a finer subdivision of an ordinal? I realize that the changes in molecular and cellular organization of the colon has been shown to be a continuum, but the subdivision of the colon into broad categories (cecum, ascending, transverse, etc) in this study is an ordinal treatment. If one could define a location in the colon reproducibly across persons using molecular markers, I think this argument could be made.
Thank you for this question. You are correct; our analysis contains a further subdivision of the colorectal frame into segments rather than using the classifier sidedness. As using the word “continuous” to describe this further subdivision could be misleading, we changed some wordings in the manuscript to reflect this consideration.
Lines are listed as they are seen with activated tracking mode.
37-40,
48,
78,
219,
222-223,
259-260, 281-282, 305
Major comments:
The authors mention that MMR status was not controlled for in this study and may have an impact, and I wholeheartedly agree with this statement.
I think, however, the bigger problem by far is the lack of controlling for morphological subtype, grade and stage of tumor. Presumably, there is an even distribution amongst the various sites, but we have no way of assessing this from the provided data. All we know is that the included patients are presumably pTx Nx M1 (e.g they have distant metastases). What if, for instance, the cecal cohort cases are enriched for patients with Grade 3, Undifferentiated, pT4 N2b M1? This is probably not the case, but it cannot be assessed based on the data provided.
Without this data, all I can say is that the findings are interesting but need further investigation.
Thank you very much for your interesting train of thought.
Grade, morphological subtype and stage of tumor were not assessed in the analysed studies, as they do not influence treatment strategy in the metastatic setting. As our pooled analysis is limited by its retrospective design, we agree that there could be undetected bias - as already acknowledged in the manuscript, e.g. in lines 313-317 – and that further studies are necessary to confirm our results – already acknowledged in lines 338-340.
As we deem your concrete concern as relevant and valuable, we addressed it as follows:
Page 14, lines 318-321 (in tracking mode):
Furthermore, additional information including exact stage (size of primary tumor, nodal status), grade and morphological subtype (e.g. well or poorly differentiated tumors) is unavailable and a potential uneven distribution of these features between segments may have biased our observation.
318-321
Reviewer 2 Report
Annabel HS Alig et al. show a robust pooled analysis of five randomized trials (FIRE-1, CIOX, FIRE-3, XELAVIRI, VOLFI) focusing on the prognostic/predictive role of primary tumour location in RAS/BRAF wild-type metastatic colorectal cancer (mCRC) patients in relation to efficacy of anti-EGFR therapy. The study is pleasant to read, well written and logically presented. Statistical methods are adequately described and applied. The discussion is stimulating and complete particularly in addressing methodological limits and scientific suggestions of the paper. The merit of this very interesting and correct article is not only the clinical pragmatic information (e.g. the detrimental effect of anti-EGFR agents in caecal localizations) but principally the further breaking of a paradigm on the biologic behaviour and associated prognostic (cancer biology surrogates necessarily prognosis) implications of sidedness (right vs. left): the prognostic power of sidedness works in a continuous rather than a dichotomous manner in RAS/BRAF wild-type mCRC patients. In my opinion, this study provides and also prompts evidence on the prognostic importance of sidedness which should impact on stratification rules in randomized trials and clinical/pathologic classifications (particularly in “proximal” right sided CRCs). I have just found a few typos that can be corrected through a minor proofreading.
Author Response
Dear reviewer,
Thank you very much for your feedback and assessment. We are very grateful for your time and your efforts.
We corrected typos throughout the manuscript.
Reviewer 3 Report
The authors revealed the primary tumor sidedness shows prognostic and predictive impact on anti-EGFR agent efficacy and thus management of metastatic colorectal cancer using the five randomized 1st-line therapy studies containing various therapy protocols concerning primary tumor location, dividing the colorectal frame into six segments. As a result, this cohort study showed the primary tumor location was associated with metastatic spread: liver, lung, peritoneal and lymph nodes significantly. Multivariate analysis indicated an impact on anti-EGFR agent efficacy in terms of overall survival depending on the exact primary tumor location: from detriment in caecal, rather neutral effects in ascending colon, right flexure/transverse colon and left flexure/descending colon to clear benefit in sigmoid and rectal primaries. This finding challenges the current use of the dichotomous left- vs. right-classifier to choose first-line treatment and may provide a basis for differentiated clinical decision-making.
This paper is useful for the readers of Cancers since it is concise and has lucid explanation and table.
Round 2
Reviewer 1 Report
The authors have adequately addressed my concerns.